# Unequal by the gun: Four decades of the Black-White firearm homicide gap

Alex Knorre[1][*], John MacDonald[2]

**1** University of South Florida, Department of Criminology, Tampa, Florida, United States of America,
**2** University of Pennsylvania, Department of Criminology, Philadelphia, Pennsylvania, United States of America

* knorre@usf.edu

## Abstract

**Background:** Firearm homicide is a leading cause of death in the United States, with substantial racial disparities documented over time. Black Americans experience disproportionately higher rates of firearm homicide compared to White Americans, reflecting long-standing social, economic, and structural inequities. This study contributes to understanding the magnitude and trends of this disparity.

**Objective:** We investigate the Black-White population-level disparity in firearm homicide rates over 45 years in the United States and estimate the fraction of excess deaths in the Black population attributable to firearm homicides.

**Methods:** We measure racial disparities in firearm homicides as the ratio of age-adjusted rates between Black and White populations. Using annual national firearm homicide and population data from CDC WONDER (1979–2023), disaggregated by race, age, and sex, we calculate age-adjusted Black–White mortality ratios by year and sex, as well as the number of excess Black firearm homicide deaths attributable to these disparities.

**Results:** After a steady period of 1990-2010, the Black-White disparity began increasing and peaked in 2020 when the firearm homicide rate in males was 10.38 times higher in Blacks than Whites. There were 31,202 excess firearm homicides of Black people attributed to the Black-White racial disparity in 2021-2023.

**Conclusions:** Despite significant declines in firearm homicides during the 1990s, the relative racial gap in firearm homicide has persisted and grown, reflecting the unequal burden of violence borne by the Black population during both declines and surges of violence.

## Introduction

Serious crime and violence in the United States is spatially concentrated in the most socially and economically disadvantaged urban neighborhoods where many residents are Black or Hispanic [1,2]. The concentration of serious crime and violence patterns is associated with long-standing racial inequalities in housing, education,

**Data availability statement:** The replication package containing data and code is available here: https://github.com/alexeyknorre/racial_gap_gun_homicides.

**Funding:** The author(s) received no specific funding for this work.

**Competing interests:** The authors have declared that no competing interests exist.

finance, criminal justice, health care, employment, and other institutions. Scholars identify these key structural forces as the main drivers of racial gaps in mortality and life expectancy in the United States, noting that they reflect evidence of structural racism [3–8]. Recent research shows that the surge in gun violence during the COVID-19 pandemic was disproportionately concentrated in high-poverty neighborhoods, where the majority of residents are Black or Hispanic, communities that already had a higher share of firearm violence before the pandemic. For example, in cities Philadelphia, New York, and Los Angeles, between 36% and 55% of the increase in shootings between 2020 and 2021 occurred in just 10% of city census block groups — areas where the residents were overwhelmingly Black and Hispanic [9]. The toll of gun violence is especially concentrated in the most affected communities. In the most economically disadvantaged neighborhoods of Chicago and Philadelphia, young males face an annual risk of gun injury or death greater than 5%, comparable to or exceeding combat risks of the US service members deployed to Afghanistan and Iraq [10].

To Since 2015, life expectancy in the U.S. has plateaued and modestly declined, with mortality increasing most notably among individuals ages 15 to 44 — a trend that reflects an enduring and ethnic inequality [11]. Between 2000 and 2010, the Black-White life expectancy gap had been narrowing, largely due to shifts in the age composition of these populations, which led to a reduction in mortality differences from chronic diseases [12]. More recently, however, the Black-White life expectancy gap has increased as a result of emerging causes, including drug overdoses, gun violence, and suicides [13].

Homicide is a major driver of the Black-White mortality gap, disproportionately affecting Black individuals, especially young adults aged 15-44 [14]. In 2016, the homicide mortality for Black males was 20 times higher than for White males in the United States [5]. Eliminating homicide mortality for Black males would result in 0.53 years of life gained, considerably larger than the effect of removing mortality from suicides (0.16), auto accidents (0.21), and drug overdoses (0.10). By contrast, the elimination of homicide mortality for White males would increase life expectancy by 0.13 years, a smaller estimate than suicides (0.24), auto accidents (0.24), and drug overdoses (0.24) [5].

Firearms are the most significant contributor to homicides among youth and young adults in the U.S., accounting for approximately 81% of all cases [15]. Firearm homicides are the major contributor to racially disparate homicide rates [16]. The Black-White inequality in firearm homicides has been documented across decades in the United States [17–19]. During 2018-2022, there was a stable Black-White gap in firearm homicide rates [20,21], which exceeded all other types of violent victimization. In 2020 and 2021, homicide was the leading contributor to disparities in life expectancy between Black and White males in the U.S., exceeding that of deaths from COVID-19 [22]. However, less is known about how Black-White inequality in firearm homicides has changed over time, particularly across periods of surges and declines in gun violence.

Racial disparities in firearm mortality rates closely correspond to surges in gun violence cycles in the U.S. Over the past three decades, the U.S. has experienced

several such cycles. In the late 1980s, there was a major increase in homicide rates associated with the introduction of crack cocaine and its attendant drug markets [23]. More youth engaged in the illegal drug trade started to carry firearms for protection, which naturally led to increased rates of firearm homicides among these age groups [17]. Starting from the mid-1990s, homicide rates began to plummet [24]. The drop in homicide rates was unexpected, and scholars from different fields speculated on the causes [25]. The decline in the homicide rates in the U.S. in the 1990s did not equally impact the same groups that experienced the rise. The racial inequality in homicides caused by the growth of homicide victimization of young Black males remained persistently higher than it was before the epidemic rise [26]. While a variety of theories have been proposed to explain the unexpected decrease in homicides in the 1990s, one of the main outcomes of the drop was the dramatic gain in the life expectancy of Black males that lasted until the 2010s [27,28].

The homicide rate in the U.S. began rising from a low point in 2014. Between 2014 and 2020, the firearm homicide rate across 100 major U.S. cities increased by 76% [29]. The rise in violence was most acute in majority-Black neighborhoods, where firearm homicides rose by 87% during this period [29]. In Chicago, for example, this increase in homicide erased the progress in Black-White inequality that had been achieved during the long decline in violence from the early 1990s to the mid-2010s. The upward trend peaked in 2020, marked by a 25% increase in the national homicide rate—the largest single-year rise ever recorded [30]. The causes of the rise in homicides since 2014 remain largely unexplored, especially given that the trend predates the pandemic-era spike. Given that firearm homicide mortality is the leading cause of death for young Black males aged 15-34 [31], it is important to understand changes in the Black-White inequality in firearm homicide rates over the longer term and during periods of surges.

This study provides new statistical evidence of excess deaths from firearm homicides in the U.S., highlighting their role in the Black–White mortality disparity over the past 45 years. We calculate the mortality ratio over 1979-2023 and estimate the fraction of excess deaths in the Black population due to the Black-White inequality in firearm homicides. Several studies have analyzed racial disparity in firearm homicides at the sub-national level in the U.S. and found robust evidence of the Black-White gap [32–35].

We follow an approach used by Preston and Vierboom [36] to estimate the age-adjusted mortality in the U.S. relative to five European countries, which has been used in several other recent applications [37–40]. Preston and Vierboom estimate that the U.S. would have had 400,700 fewer deaths between 2000 and 2017 if it had a similar mortality rate as European comparisons. In this paper, we follow the same approach and examine the Black-White ratio of firearm homicide mortality rates between 1979 and 2023 after adjusting for sex, age, and calendar year effects (hereafter, we use "years" to refer to calendar years for brevity). We rely on these comparisons to calculate the attributable fraction of excess firearm homicide deaths for the Black population.

## Materials and methods

### Data

We obtained data from the Wide-ranging Online Data for Epidemiological Research (WONDER) database maintained by the Centers for Disease Control and Prevention (CDC). We accessed and compiled firearm homicides from three sources that provide details on firearm homicides in the US: Compressed Mortality (1979-1998), Compressed Mortality (1999-2016), and Underlying Cause of Death (2017-2023). From these files, we selected Homicide as the Injury Intent and Firearm as the Injury Mechanism. We obtained firearm homicide counts and populations for every single year, race, age group, and sex at the national level. We did not have access to personally identifiable information and compiled these publicly available datasets in aggregated form on December 1, 2022 and then accessed new years of data on June 15, 2025. The replication package containing data and code is available here: https://github.com/alexeyknorre/racial_gap_gun_homicides

One significant limitation of the data from 1979-1998 is that it only contains records on three racial categories (Blacks, Whites, and "Other"), so we focus our analysis on comparing Black and White populations only. We aggregated age

groups into 5-year (10-14, 15-19, 20-24) and 10-year (0-9, 25-34, 35-44,…, 75-84) intervals to ensure comparability across CDC databases. The age group of 85 or older was removed because it is often suppressed due to low counts.

Another limitation of the data is the transition from bridged-race categories to single-race categories for Underlying Cause of Death (UCD) data spanning 1999-2023. In 2021, the CDC stopped providing data with the four mutually exclusive bridged race-ethnicity categories: White, Black, Asian, and American Indian. Instead, starting in 2018, CDC WONDER rolled out a more granular measure of race and ethnicity (single race categories), with either 6, 15, or 31 racial categories. Specifically, UCD data with 6 race categories contain Black, White, Asian, American Indian, Pacific Islander, and "More than one race". This breaks compatibility with the pre-2021 UCD data because now some people can be categorized as "More than one race", thus deflating death and population counts, particularly for Black individuals. Thus, UCD-6 cannot be used to recreate race-specific mortality rates for years after 2020 that are fully consistent with previous years. UCD is also available with more granular racial and ethnic categories in two forms, containing 15 and 31 mutually exclusive options. However, UCD with more than 6 race and ethnicity categories does not provide total population numbers.

To address the change in data categorizations, we proceed with our analysis by combining UCD's four bridged-race categories data in 2018-2020 and six single-race categories data in 2018-2023. Single-race categories are created in two possible variations: with and without adding "More than one race" records to Black and White population and death counts. While this does not allow us to perfectly align post-2020 estimates with previous years, it creates upper and lower bounds for the change in 2021-2023, allowing us to see the direction of change in the racial gap in homicide mortality from firearms. Importantly, the change in CDC's race and ethnicity categories does not affect the estimates of excess deaths, as can be seen from Fig 4.

Our final analytic sample contains a total of 607,315 firearm homicides of Black and White victims that occurred in the U.S. from 1979 to 2023. This number excludes 15,812 firearm homicides with the race coded as "Other" before 1999. The final analytic dataset includes deaths for combined Single and Multiple race categories recorded in 2018-2023 but excludes deaths for Single-race categories.

## Methods

**Rate and ratio calculation.** We calculated the firearm homicide mortality rate (DR) by dividing the total number of firearm homicides by the population for each age group $a$, year $y$, sex $s$, and race $r$:

$$DR_{a,y,s,r} = \frac{\text{Firearm homicide deaths}_{a,y,s,r}}{\text{Population}_{a,y,s,r}} \times 100{,}000 \tag{1}$$

We then calculated the mortality ratio (MR) by dividing the rates for the Black population by the rates for the White population at each subgroup of age, year, and sex:

$$MR_{a,y,s} = \frac{DR_{a,y,s|r=Black}}{DR_{a,y,s|r=White}} \tag{2}$$

For each year and sex, we then calculated the age-adjusted mortality ratio. To do this, we calculated the age-adjusted mortality rates using the direct method [41] which accounts for the differences in the underlying age structure of the Black and White populations. We used the age population structure separately for males and females as the standard at each given year according to the following calculation:

$$AADR_{y,s|r=White} = \sum_a w_{a,y,s} \times DR_{a=i,y,s|r=White} \tag{3}$$

where $w_{a,y,s}$ is the weight given by the referential age population structure of Blacks for a given year and sex as in the following calculation:

$$w_{a,y,s} = \frac{\text{Population}_{a,y,s|r=Black}}{\sum \text{Population}_{a,y,s|r=Black}} \tag{4}$$

The age-adjusted firearm homicide mortality ratio (AAMR) for each year and sex is given by the final calculation shown in Eq 5:

$$AAMR_{y,s} = \frac{DR_{y,s|r=Black}}{AADR_{y,s|r=White}} \tag{5}$$

The resulting age-adjusted mortality ratio estimates the relative Black-White racial gap in firearm homicide mortality. The resulting number shows how much more likely a Black person is to die from a firearm homicide than a White person of the same age group, sex, and in the same year.

**Excess deaths.** We estimated the number of excess deaths resulting from this gap. Following the approach of [36], the number of excess Black firearm homicide victims (ED) for a given age group $a$, year $y$ and sex $s$ is calculated as the difference between the actual number of firearm deaths within the subgroup and the counterfactual number of deaths that would have occurred if the Black population had the same age, sex, and year-specific firearm homicide rate as the White population:

$$ED_{a,y,s} = \underbrace{Deaths_{a,y,s|r=Black}}_{\text{Actual deaths}} \tag{6}$$
$$- \underbrace{DR_{a,y,s|r=White} \times 100,000^{-1} \times \text{Population}_{a,y,s|r=Black}}_{\text{Counterfactual deaths}}$$

We plot the total number of excess deaths attributable to firearm homicides in the Black population over time. For clarity and parsimony, we present combined estimates of excess deaths by age and sex in three-year increments. Using three-year periods helps smooth out year-to-year fluctuations and allows for more stable and interpretable trend estimates, especially given the year-to-year variability in firearm homicides. We focus on the three-year periods with the highest number of Black excess deaths between 1979 and 2023, highlighting peaks in the early 1990s (1991–1993), during the post-2014 rise (2018–2020), and in the most recent years of available data (2021–2023).

## Results

### Racial gap in firearm homicide mortality

Fig 1 shows the crude firearm homicide mortality rates since 1979 for Black and White males and females. Since 1978, there have been three major spikes in firearm homicides in the U.S. that occurred in 1980, 1993, and 2020. The highest rate of 7.20 firearm homicides per 100,000 people occurred in 1993, while there were a total of 6.36 firearm homicides per 100,000 in 2020. The burden of firearm homicides is disproportionately concentrated among Black males, with Black females having a rate of firearm homicides similar to that of White males.

As previously explained, the CDC changed the classification of race in 2021, transitioning from bridged-race to single/multiple-race categories. We show three different race trends in our plots to account for the changing racial classification, yet the results differ only slightly.

Fig 2 presents the main results with the age-adjusted firearm mortality ratio between Black and White males and females, along with the 95% confidence interval. There was relative stability in the Black-White male inequality in firearm homicides between 1988 and 2010. Black males were on average eight to nine times more likely than White males to die from a firearm homicide. Starting in 2010, the Black-White inequality started to grow slowly, even though this period of

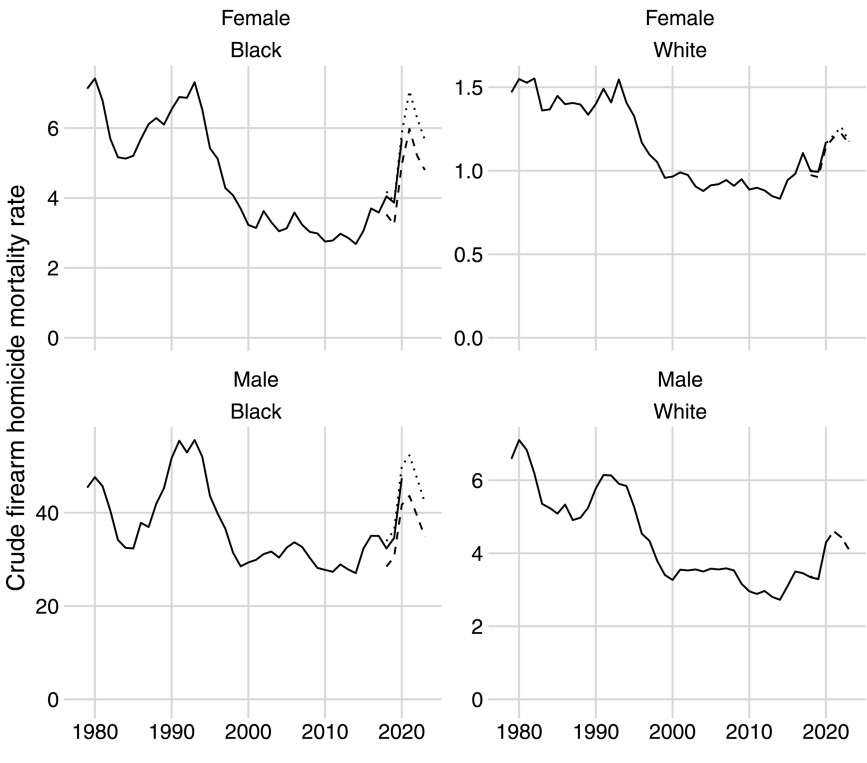

**Fig 1**. **Crude firearm homicide rates in the United States in 1979-2023.** Rates are not adjusted for age and represent counts of firearm homicides per 100,000 population.

2010-2015 had the lowest absolute rate of firearm homicides of Black males. In 2020 the Black-White male gap in firearm homicides was at its highest since 1979, with Black males being 10.38 times more likely than White males to die from a firearm homicide. If we use single-race categories instead of bridged-race categories, the racial gap increases to 10.88. In 2021-2023, this racial gap decreased only slightly.

The historical change in the Black-White gap of firearm mortality for females is significantly smaller than for males. The onset of racial disparity in lethal shootings for females started in 1979, with Black females being 4.96 times more likely than White females to be the victim of firearm homicide. The Black-White gap in firearm homicides for females narrowed between 1979 and 2017. Since 2017, the Black-White inequality in female firearm homicide rates has grown, reached a peak of 4.74 in 2020, continued to grow in 2021, and has since decreased similarly to the male racial gap between 2021 and 2023.

Fig 3 further subsets the analysis by age groups. Over the last two decades, the Black-White gap has been increasing among young males. Compared to the era of the high violence of 1985-1994, Black males aged 25-34 now have much higher rates of mortality from firearm homicides than in the early 1990s when homicide rates were at peak levels.

**Excess deaths**

Fig 4 shows how the Black-White gap in firearm homicide mortality translates into the number of deaths that would not have occurred if this gap disappeared each year. Table 1 shows these calculations by sex, age, and race for three-year increments of 1991-1993, 2018-2020, and 2021-2023. In 1991-1993, 28,254 of the victims of firearm homicides were

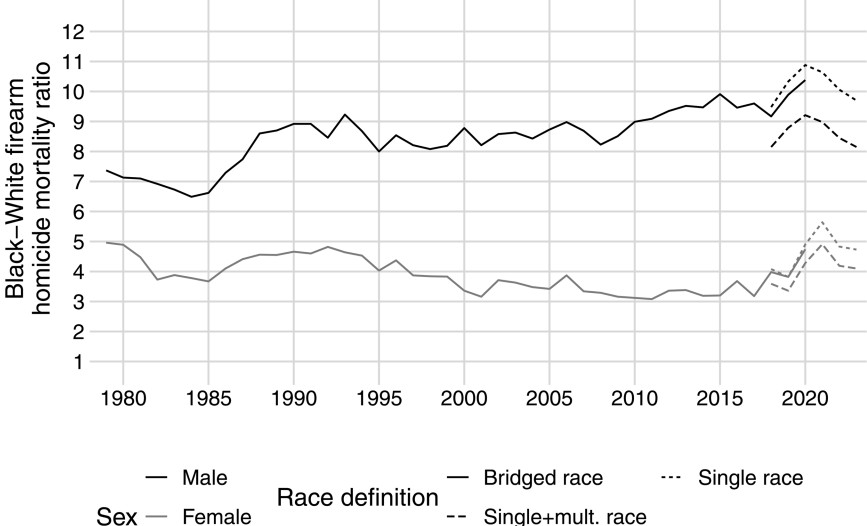

**Fig 2**. **Changing racial gap in the age-adjusted firearm homicide mortality ratio.** The lines show yearly mortality ratios of age-adjusted Black-White firearm homicide rates. The grey area shows 95% confidence intervals.

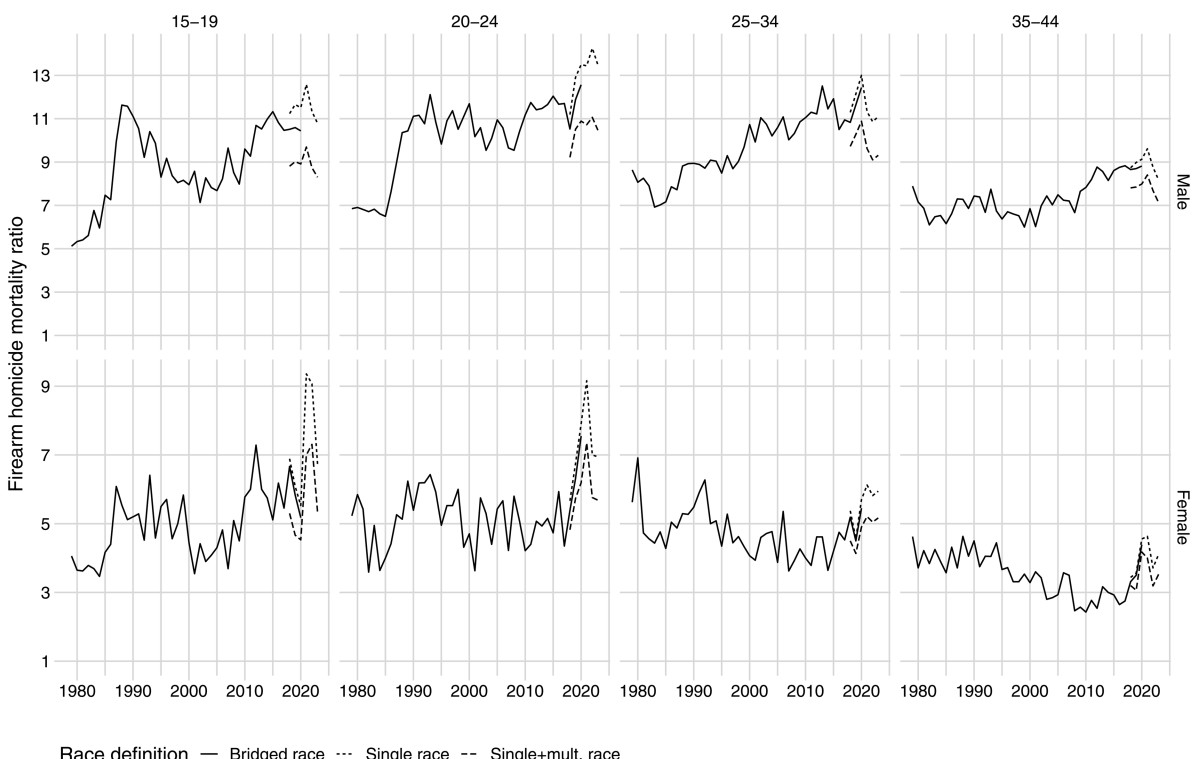

**Fig 3**. **Black-White firearm homicide mortality gap by age groups.**

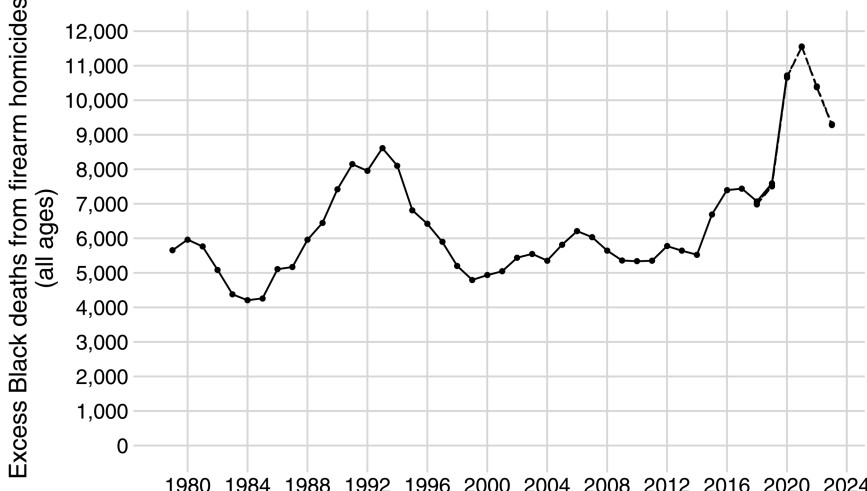

Race definition — Bridged race ··· Single race -- Single+mult. race

**Fig 4**. **Excess Black firearm deaths relative to White over time.**

**Table 1**. **Excess Firearm Homicide of Black Population.**

| | Firearm homicide deaths | | | | | | | | |
| | 1991-93 | | | 2018-20 | | | 2021-23 | | |
| Age | Actual | Counterfactual | Excess | Actual | Counterfactual | Excess | Actual | Counterfactual | Excess |
|---|---|---|---|---|---|---|---|---|---|
| 0-9 | 187 | 49 | 138 | 226 | 41 | 185 | 265 | 47 | 218 |
| 10-14 | 502 | 87 | 415 | 251 | 50 | 201 | 500 | 81 | 419 |
| 15-19 | 5,459 | 585 | 4,874 | 3,789 | 390 | 3,399 | 5,303 | 621 | 4,682 |
| 20-24 | 7,316 | 696 | 6,620 | 5,960 | 552 | 5,408 | 6,824 | 688 | 6,136 |
| 25-34 | 8,626 | 1,047 | 7,579 | 10,029 | 965 | 9,064 | 11,736 | 1,375 | 10,361 |
| 35-44 | 3,956 | 614 | 3,342 | 4,924 | 652 | 4,272 | 6,484 | 952 | 5,532 |
| 45-54 | 1,310 | 267 | 1,043 | 2,101 | 376 | 1,725 | 2,874 | 516 | 2,358 |
| 55-64 | 518 | 117 | 401 | 1,083 | 216 | 867 | 1,440 | 276 | 1,164 |
| 65-74 | 293 | 56 | 237 | 309 | 81 | 228 | 437 | 109 | 328 |
| 75-84 | 87 | 22 | 65 | 60 | 34 | 26 | 42 | 38 | 4 |
| Total | 28,254 | 3,540 | 24,714 | 28,732 | 3,357 | 25,375 | 35,905 | 4,703 | 31,202 |

Columns "Actual" show counts of Black firearm homicides by age groups and year periods. Columns "Counterfactual" show estimated counts of Black firearm homicides if they had experienced White firearm homicide rates for the respective age group and year period. Columns "Excess" show the difference between actual and counterfactual counts.

Black. If the Black population experienced age and sex-specific mortality rates similar to the White population, we project that there would be 3,540 Black victims of firearm homicides during this period. After taking the difference between the two, there is an estimated 24,714 excess deaths solely attributable to the relative Black-White inequality in firearm homicide mortality.

A similar calculation for 2018-2020 period, where 2020 was the year with the highest absolute rate of 7 firearm homicide deaths per 100,000 in the United States, yields 25,375 excess Black firearm homicides. In 2021-2023, there were an estimated 31,202 excess Black firearm homicides. Together, these estimates indicate that the relative inequality in firearm

homicide deaths affecting the Black population in the United States has grown substantially since 2014, surpassing levels seen even in the early 1990s, with the period from 2016 to 2023 standing out as the most severe on record when age-adjusted comparisons to the White population are taken into account. As can be seen from Table 1, excess firearm homicides of Black youth (age 10-24) are roughly similar across cohorts of 1991-1993 and 2021-2023. The increase in excess Black firearm homicides in 2021-2023 compared to 1991-1993 can be attributed to the age groups of 25 years and older.

## Discussion

The great homicide decline in the United States in the 1990s has been celebrated as a public health achievement [27]. Although the absolute rates of firearm homicides decreased between the mid-1990s and mid-2010s for the Black population, the Black-White inequality in firearm homicide rates did not [14]. The ratio of Black-to-White male firearm homicide rates held steady at about 8 or 9 between 1990 and 2010. However, it began to climb after 2010, peaking at 10.38 in 2020. This growth in homicide since 2014 caused by firearms has erased the benefits of the great homicide decline of the 1990 to 2010 period for the Black population in the United States. This disparity results in thousands of excess lives lost among the Black population in the U.S. In 2021 alone, we estimate there were 11,548 excess Black victims of firearm homicides, a number that is even higher than that estimated for 1993 when the U.S. had the highest recorded firearm homicide rate. The period from 2018 to 2023 saw the highest number of excess firearm homicides among Black Americans compared to any other six-year span since 1979.

While the 2020 spike in firearm homicides is a topic that has received national attention, this surge masks a longer trend of increasing Black-White inequality in firearm homicides [42]. Many studies in the social sciences and public health have focused on the concentrated poverty and other social disadvantages in predominantly Black neighborhoods and its association with violence and health outcomes [27,43,44]. The long-standing racial disparities in violent and firearm victimization in the U.S. are tied to other dimensions of structural inequality, and suggest action is needed to address these enduring differences by race [3–7]. At the same time, there is no clear evidence for causal pathways between the rise in homicide and the growth in the Black-White inequality in firearm homicides that started in 2015 and accelerated during the COVID-19 pandemic. Rosenfeld argues that the recent upturn in homicide and violent crime started in 2015 and can be explained by two mechanisms [45]. The first is an overall reduction in policing effectiveness stemming from the reduced proactive enforcement activity and compromised police legitimacy, with racial minority communities being affected the most. The second explanation for the increase in homicide and violent crime is the opioid epidemic and its impact on illegal drug markets. While the opioid epidemic has affected large swaths of the rural White population [46], its connection to the dramatic racial gap in firearm homicides is unclear.

The analysis presented here has several limitations. First, we only look at Black and White firearm homicide rates due to the unavailability of a more granular indication of race and ethnicity. A more advanced study would specifically look at firearm homicide mortality among other racial and ethnic groups, particularly Hispanics. Second, given that the Black population in the U.S. is experiencing disproportionately more concentrated disadvantage, further studies might look at the long-term trends of firearm homicides while simultaneously controlling for socioeconomic characteristics of neighborhoods. An extensive literature already documents these trends at the level of states, counties, and selected cities [32–35]. More research is needed to understand how local contexts and policies are associated with the racial gap in firearms homicide. Policy evaluations and quasi-experimental studies are required to identify which interventions are effective at reducing these disparities over time.

High levels of firearm homicide victimization in the Black population in the United States appear to be strongly implicated in population-level life expectancy [22]. This study demonstrates that during surges in gun violence in the United States, the Black population experiences thousands of excess deaths that could be avoided if public health and policy efforts were effective at reducing the racial gap in homicide victimization for people living in the U.S.

## Author contributions

**Conceptualization:** Alex Knorre, John MacDonald.

**Data curation:** Alex Knorre, John MacDonald.

**Formal analysis:** Alex Knorre, John MacDonald.

**Funding acquisition:** Alex Knorre, John MacDonald.

**Investigation:** Alex Knorre, John MacDonald.

**Methodology:** Alex Knorre, John MacDonald.

**Project administration:** Alex Knorre, John MacDonald.

**Resources:** Alex Knorre, John MacDonald.

**Software:** Alex Knorre, John MacDonald.

**Supervision:** Alex Knorre, John MacDonald.

**Validation:** Alex Knorre, John MacDonald.

**Visualization:** Alex Knorre, John MacDonald.

**Writing – original draft:** Alex Knorre, John MacDonald.

**Writing – review & editing:** Alex Knorre, John MacDonald.

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
