## [Decision Letter · Decision Letter 0]

14 Aug 2025

PONE-D-25-38735Unequal by the gun: Four decades of the Black-White firearm homicide gapPLOS ONE

Dear Dr. Knorre,

Thank you for submitting your manuscript to PLOS ONE. After careful consideration, we feel that it has merit but does not fully meet PLOS ONE’s publication criteria as it currently stands. Therefore, we invite you to submit a revised version of the manuscript that addresses the points raised during the review process.

**ACADEMIC EDITOR:**

The peer-review process has been completed. The reviewers have requested some major issues that need to be addressed before the manuscript can be considered for publication. The detailed feedback from reviewers is included below/attached for your reference.

We kindly request that you address these points in your revised manuscript and provide a response letter detailing the changes made. Please submit the revised version of your manuscript along with the response letter through our submission system.

We look forward to receiving your revised manuscript.

We look forward to receiving your revised manuscript.

Kind regards,

Claudio Alberto Dávila-Cervantes, Ph.D.

Academic Editor

PLOS ONE

Journal Requirements:

“Alex Knorre acknowledges that this work is supported in part by funds from the AFOSR MURI grant #FA9550-22-1-0380.”

“Alex Knorre acknowledges that this work is supported in part by funds from the AFOSR MURI grant #FA9550-22-1-0380.”

“Alex Knorre acknowledges that this work is supported in part by funds from the AFOSR MURI grant #FA9550-22-1-0380.”

“The author(s) received no specific funding for this work”

7. When completing the data availability statement of the submission form, you indicated that you will make your data available on acceptance. We strongly recommend all authors decide on a data sharing plan before acceptance, as the process can be lengthy and hold up publication timelines. Please note that, though access restrictions are acceptable now, your entire data will need to be made freely accessible if your manuscript is accepted for publication. This policy applies to all data except where public deposition would breach compliance with the protocol approved by your research ethics board. If you are unable to adhere to our open data policy, please kindly revise your statement to explain your reasoning and we will seek the editor's input on an exemption. Please be assured that, once you have provided your new statement, the assessment of your exemption will not hold up the peer review process.

Reviewers' comments:

Reviewer's Responses to Questions

**Comments to the Author**

1. Is the manuscript technically sound, and do the data support the conclusions?

Reviewer #1: Yes

Reviewer #2: Yes

Reviewer #3: Yes

2. Has the statistical analysis been performed appropriately and rigorously?

Reviewer #1: Yes

Reviewer #2: Yes

Reviewer #3: No

3. Have the authors made all data underlying the findings in their manuscript fully available?

Reviewer #1: Yes

Reviewer #2: Yes

Reviewer #3: Yes

4. Is the manuscript presented in an intelligible fashion and written in standard English?

Reviewer #1: Yes

Reviewer #2: Yes

Reviewer #3: Yes

5. Review Comments to the Author

Reviewer #1: MAJOR

Abstract: The abstract leaves a lot to be desired:

• missing “Background” “Methods” “Results” and “Conclusion” sections.

• the data are national for the U.S.

• that there are two calendar-year inflections in increases in the White:Black comparison (“mid-1980s” and 2014).

• that several years had greatest Black vs. White disparity (2020, 2021 and 2023).

• that 31,202 homicides cited as the disparity without indicating over what years and geography (instead expressing the difference with age-adjusted rates).

• “Black men were 10.38 times more likely than White men to die from firearm homicide” can be interpreted to mean that a homicide attempt was 10x more likely to result in death.

• “The firearm homicide rate in males was 10 times higher in Blacks than Whites” would be more accurate. [a 2 decimal number is not necessary].

• The “number of excess Black firearm homicide deaths” should be “rate of excess …” and “homicide deaths” should be “homicides.”

Methods: Superbly explained.

Results: In general, adequately presented.

Discussion: In general, adequately presented. Not included as potential causes are firearm theft rates differences, varying state regulations in firearm purchasing and ownership, firearm industry policies regarding sales racial pitches, and possibly mass shooting differences in Blacks and Whites.

MINOR

In general, “women” should be “females” and “men” (used 19 times) should be “males.”

“100 thousand” should be “100,000”.

Introduction:

• What is meant by “credit” and “other institutions” among the racial inequalities?

• What does “top 10% of block groups” mean?

• How does the reader compare the “greater than 5%” gun injury or death rate in Chicago and Philadelphia with other cities and rural settings if comparative values are not provided?

• “in the decade prior” dangles without specifying the actual years

• How do “shifts in the age composition” explain the life-expectancy gap between Whites and Blacks?

• “In 2016, homicide mortality for Black males was 20 times higher than for White males.” In the U.S.?

• “`Firearms are the most significant contributor to homicides among youth and young adults in the U.S., with firearms accounting for approximately 80.5% of all homicides.” 80.5% is quite finite and not “approximate”.

• “”… we follow a similar design and examine the Black-White ratio of firearm homicide mortality rates between 1979 and 2023 after adjusting for sex, age, and year effects.” Since age is in years, “year” should probably be “calendar year”. Other uses of “year’ in rest of manuscript should probably specific “calendar year”.

Table 1 title/legend lacks description of “Counterfactual”.

Line 172: “similar’ would be more appropriate than “same”.

Lines 173-174: “Excluding the period of 1989-1993, the decrease in firearm homicide rates for Black males is less pronounced than for other groups” is a qualitive statement that is difficult to understand. If anything, the changes after 2000 suggest that Black and White males had similar trends.

Line 207: delete “there’.

Table 1: From 1991-1993 to 2018-2020, the excess number of firearm homicides in Blacks decreased in 10-23 year-old, with a total decrease in excess deaths of 2,901. That young Blacks were spared the overall increase from 1991-1993 to 2018-2020 should be recognized and potentially discussed, albeit it is <10% of the overall excess (of all ages).

Figure 3: Ideally the White and Black data should be shown on the same graph and given the overlap would only be for 2021-2022 for the single race data of ages 15-19 and 20-24, this option is feasible.

Figure 4: Does not specify age range.

Acknowledgements: Why does only one of the two authors acknowledge the source of financial support?

References: Their formatting and content is inconsistent, with doi in some an not others and some titles capitalized and in others not. Internet citations lack date of access. #13 had two doi citations. #9 and #18 repeats the e number. #23, #24, #37 and #40 appear to be incomplete.

Reviewer #2: This is an excellent paper that documents the racial disparities in firearm victimization over time and shows the three different peak differences where the # of excess deaths among blacks is staggering. In this regard, the paper nicely improves some recent work by Piquero and Roman, also using the CDC Wonder data, but does so by looking at longer time periods, dealing with the racial category changes, and calculating excess deaths. I have four comments for the authors to consider.

1. By my calculation, the total # of excess deaths in the thee key peak periods that the authors look at is 81,291. I think it may be helpful for the authors to contextualize this. One way to do this is the following. The Dallas Cowboys NFL Stadium holds around 80,000 people (can be expanded for standing room only). Taking the # of excess deaths means that all those deceased persons can fit in the stadium. There likely is a better way to say this but hopefully the authors get the point. That kind of visual, however depressing, is powerful.

2. I know the authors did not do anything w/ Hispanic ethnicity (though they could look at it in more recent times). Could they do a subset analysis of Hispanic ethnicity (I am not requiring it), but if not at least comment on it.

3. The paper lends heavily on structural explanations of the differential risk of firearm victimization. I do not discount that this matters, however, people still have to pick up guns AND use them. There still remains some sort of individual-level decision making here, whether its code of the street, self-control, etc. I think the authors need to point that out.

4. I like descriptive papers. A lot. Yet, not much policy commentary is here. I think a sentence or two would be useful.

Reviewer #3: Overall Comments:

The authors of this manuscript examine trends in Black–White disparities in firearm homicide over four decades. While such disparities are well-documented in the literature, this study contributes by providing more precise estimates—specifically, ratios quantifying the disparity—and by calculating measures of excess mortality.

However, I am concerned by the absence of a substantive investigation into racism as a driver of these trends. The framing largely gives a cursory mention of socioeconomic and neighborhood-level characteristics without adequately addressing the structural and systemic racism that underlies and shapes these factors. Without explicitly engaging with racism as a root cause, the analysis risks reinforcing race as a proxy for other factors rather than interrogating the historical and ongoing policies, practices, and power structures that produce and maintain these disparities.

Integrating a discussion of racism—at minimum through a theoretical lens, but ideally via an empirical consideration—would strengthen the manuscript, situating the observed trends within the broader social and political context that creates disparities in firearm homicide.

Introduction:

1. This section is a bit lengthy and redundant in places. For example, the authors include a lengthy discussion the place-boundedness of these racial disparities, but since this is not the focus of the paper, this section could be reduced or moved to the discussion.

2. The introduction is missing several references to existing literature on firearm homicide disparities. These may also fit in the discussion.

a. Wong B, Bernstein S, Jay J, Siegel M. Differences in racial disparities in firearm homicide across cities: the role of racial residential segregation and gaps in structural disadvantage. Journal of the National Medical Association. 2020 Oct 1;112(5):518-30.

b. Conrick KM, Adhia A, Ellyson A, Haviland MJ, Lyons VH, Mills B, Rowhani-Rahbar A. Race, structural racism and racial disparities in firearm homicide victimisation. Injury prevention. 2023 Aug 1;29(4):290-5.

c. Siegel M, Rieders M, Rieders H, Moumneh J, Asfour J, Oh J, Oh S. Measuring structural racism and its association with racial disparities in firearm homicide. Journal of racial and ethnic health disparities. 2023 Dec;10(6):3115-30.

d. Knopov A, Rothman EF, Cronin SW, Franklin L, Cansever A, Potter F, Mesic A, Sharma A, Xuan Z, Siegel M, Hemenway D. The role of racial residential segregation in Black-White disparities in firearm homicide at the state level in the United States, 1991-2015. Journal of the National Medical Association. 2019 Feb 1;111(1):62-75.

e. Bottiani JH, Camacho DA, Lindstrom Johnson S, Bradshaw CP. Annual research review: youth firearm violence disparities in the United States and implications for prevention. Journal of child psychology and psychiatry. 2021 May;62(5):563-79.

Methods:

3. Why did the authors not conduct formal trend analyses, such as via Joinpoint? This would allow the conclusions regarding spikes to be more sound.

Results:

4. The terminology of “fatal shootings” is new in this section and could be misleading. Please specify the unit is firearm homicides per 100 thousand people. Additionally, were these truly whole numbers? That is “7.0 per 100 thousand;” if not, please include the precise numbers.

5. Please be consistent in using terms related to sex (male, female) compared to gender (men, women).

6. It would be helpful to discuss in what ways trends changed with the change in race definitions.

7. In Figure 1, the difference in the dashed line between single race and single + multi race is difficult to distinguish. Consider making one a dotted line. Additionally, the major differences in y-axes make it difficult to make visual comparisons.

8. In Figures 1-2, what is the unit for the y-axis? Per 100 thousand?

Discussion

9. The discussion needs expanded explanations for these trends. Remarkably, the word “racism” does not appear at all in the manuscript. Please see: https://www.healthaffairs.org/content/forefront/racism-new-standard-publishing-racial-health-inequities. There is an extremely large body of literature examining the association of structural racism with disparities in firearm homicide that provides several causal explanations.

10. “At the same time, there is no clear causal evidence for the rise in homicide and the growth in the Black-White inequality in firearm homicides that started in 2015 and accelerated during the COVID-19 pandemic.” This sentence is a bit misleading; I believe the authors mean to say that this literature is nascent, and we are still identifying the causal pathway. However, it may be misinterpreted to mean that there is no causal relationship at all, rather than that the current evidence base is limited and evolving. Clarifying that the body of research is emerging, and that causal mechanisms are still being explored, would help avoid confusion.

11. Regarding the section on poverty, please also see:

a. Ellyson AM, Rivara FP, Rowhani-Rahbar A. Poverty and firearm-related deaths among US youth. JAMA pediatrics. 2022 Feb 1;176(2):e214819-.

b. Barrett JT, Lee LK, Monuteaux MC, Farrell CA, Hoffmann JA, Fleegler EW. Association of county-level poverty and inequities with firearm-related mortality in US youth. JAMA pediatrics. 2022 Feb 1;176(2):e214822-.

12. Please remove the recommendation that future studies control for neighborhood socioeconomic characteristics for two reasons. First, an extensive body of literature already examines neighborhood characteristics and disparities in firearm homicide. Second, rather than controlling for these characteristics, it would be more useful for future research to identify which specific characteristics are associated with higher or lower disparities.

6. PLOS authors have the option to publish the peer review history of their article (what does this mean?). If published, this will include your full peer review and any attached files.

Reviewer #1: **Yes:**Archie Bleyer

Reviewer #2: No

Reviewer #3: No

---

## [Author Response · Author response to Decision Letter 1]

15 Sep 2025

Please see the attached response memo.

---

## [Decision Letter · Decision Letter 1]

5 Dec 2025

PONE-D-25-38735R1Unequal by the gun: Four decades of the Black-White firearm homicide gapPLOS ONE

Dear Dr. Knorre,

Thank you for submitting your manuscript to PLOS ONE. After careful consideration, we feel that it has merit but does not fully meet PLOS ONE’s publication criteria as it currently stands. Therefore, we invite you to submit a revised version of the manuscript that addresses the points raised during the review process.

We look forward to receiving your revised manuscript.

Kind regards,

Claudio Alberto Dávila-Cervantes, Ph.D.

Academic Editor

PLOS ONE

Journal Requirements:

Reviewers' comments:

Reviewer's Responses to Questions

**Comments to the Author**

1. If the authors have adequately addressed your comments raised in a previous round of review and you feel that this manuscript is now acceptable for publication, you may indicate that here to bypass the “Comments to the Author” section, enter your conflict of interest statement in the “Confidential to Editor” section, and submit your "Accept" recommendation.

Reviewer #2: All comments have been addressed

Reviewer #3: (No Response)

Reviewer #4: All comments have been addressed

2. Is the manuscript technically sound, and do the data support the conclusions?

Reviewer #2: Yes

Reviewer #3: Partly

Reviewer #4: Yes

3. Has the statistical analysis been performed appropriately and rigorously?

Reviewer #2: Yes

Reviewer #3: Yes

Reviewer #4: Yes

4. Have the authors made all data underlying the findings in their manuscript fully available?

Reviewer #2: Yes

Reviewer #3: Yes

Reviewer #4: Yes

5. Is the manuscript presented in an intelligible fashion and written in standard English?

Reviewer #2: Yes

Reviewer #3: Yes

Reviewer #4: Yes

6. Review Comments to the Author

Reviewer #2:

Reviewer #3: The authors’ response—adding a sentence citing literature that "invokes" structural racism but declining to use the term “racism” explicitly because of concerns about causal certainty—does not address the core issue. The manuscript frames socioeconomic and neighborhood-level characteristics in cursory terms without situating them within the power structures, policies, and historical practices that produce these factors. In the fields of public health, medicine, criminology, and social work, among others, structural racism is widely understood as the organizing framework through which disparities in housing, policing, economic opportunity, and access to safety are generated, even when a single dataset cannot trace every mechanism in detail. Without naming racism, the introduction and discussion risk reducing racial disparities to proxy variables rather than recognizing the root causes documented across multiple disciplines. This omission undermines the manuscript’s scientific clarity and leaves it out of step with current scientific expectations for accurately naming and addressing the structural drivers of racial health inequities, namely structural racism.

Reviewer #4: I did not review the first draft of this paper, but I believe the revised manuscript has adequately addressed the reviewers’ comments. The paper identifies important emerging patterns in U.S. firearm homicide mortality and employs methodologically sound and sophisticated analytical approaches. The racial disparities documented here have significant policy and theoretical implications, and future research will be needed to unpack the mechanisms driving these widening disparities.

It would also be helpful for the author to specify where the data and replication files will be made publicly available once the paper is published.

7. PLOS authors have the option to publish the peer review history of their article (what does this mean?). If published, this will include your full peer review and any attached files.

Reviewer #2: No

Reviewer #3: No

Reviewer #4: No

---

## [Author Response · Author response to Decision Letter 2]

10 Dec 2025

Please see the attached PDF with the response.

---

## [Decision Letter · Decision Letter 2]

11 Jan 2026

Unequal by the gun: Four decades of the Black-White firearm homicide gap

PONE-D-25-38735R2

Dear Dr. Knorre,

We’re pleased to inform you that your manuscript has been judged scientifically suitable for publication and will be formally accepted for publication once it meets all outstanding technical requirements.

Kind regards,

Annesha Sil, Ph.D.

Staff Editor

PLOS One

Additional Editor Comments (optional):

Reviewers' comments:

Reviewer's Responses to Questions

**Comments to the Author**

1. If the authors have adequately addressed your comments raised in a previous round of review and you feel that this manuscript is now acceptable for publication, you may indicate that here to bypass the “Comments to the Author” section, enter your conflict of interest statement in the “Confidential to Editor” section, and submit your "Accept" recommendation.

Reviewer #3: All comments have been addressed

Reviewer #4: All comments have been addressed

2. Is the manuscript technically sound, and do the data support the conclusions?

Reviewer #3: Yes

Reviewer #4: Yes

3. Has the statistical analysis been performed appropriately and rigorously?

Reviewer #3: Yes

Reviewer #4: Yes

4. Have the authors made all data underlying the findings in their manuscript fully available?

Reviewer #3: Yes

Reviewer #4: Yes

5. Is the manuscript presented in an intelligible fashion and written in standard English?

Reviewer #3: Yes

Reviewer #4: Yes

6. Review Comments to the Author

Reviewer #3: Thank you for the revisions and for engaging with this comment. I appreciate that the manuscript now explicitly uses the term “structural racism” at least once. While the framing remains cautious and somewhat indirect, the added language provides clearer conceptual grounding than the prior version. I have no further comments on this point.

Reviewer #4: No additional comments, the authors have properly addressed all the reviewers' comments in this submission.

7. PLOS authors have the option to publish the peer review history of their article (what does this mean?). If published, this will include your full peer review and any attached files.

Reviewer #3: No

Reviewer #4: No

---

## [Editor Report · Acceptance letter]

PONE-D-25-38735R2

PLOS One

Dear Dr. Knorre,

I'm pleased to inform you that your manuscript has been deemed suitable for publication in PLOS One. Congratulations! Your manuscript is now being handed over to our production team.

Kind regards,

on behalf of

Dr Annesha Sil

Staff Editor

PLOS One